



# Automatic quality control of telemetric rain gauge data providing quantitative quality information (RainGaugeQC)

Katarzyna Ośródka[1], Irena Otop[2], Jan Szturc[1]

[1]Centre of Meteorological Modelling, Institute of Meteorology and Water Management – National Research Institute, PL 01-673 Warszawa, Podleśna 61, Poland
[2]Research and Development Centre, Institute of Meteorology and Water Management – National Research Institute, PL 01-673 Warszawa, Podleśna 61, Poland

*Correspondence to*: Jan Szturc (jan.szturc@imgw.pl)

**Abstract.** The RainGaugeQC scheme described in this paper is intended for real-time quality control of telemetric rain gauge data. It consists of several checks: detection of exceedance of the natural limit and climate-based threshold, and checking of the conformity of rain gauge and radar observations, the consistency of time series from heated and unheated sensors, and the spatial consistency of adjacent gauges. The proposed approach is focused on assessing the reliability of individual rain gauge observations. A quantitative indicator of reliability, called the quality index (*QI*), describes the quality of each measurement as a number in the range from 0.0 (completely unreliable measurement) to 1.0 (perfect measurement). The *QI* of a measurement which fails any check is lowered, and only a measurement very likely to be erroneous is replaced with a "no data" value. The quality information provided is very useful in further applications of rain gauge data. The scheme is used operationally by the Polish national meteorological and hydrological service (Institute of Meteorology and Water Management – National Research Institute).

## 1 Introduction

The accuracy of telemetric rain gauge data is vital both for scientific research and for real-time modelling. Reliable precipitation measurements with high temporal and spatial resolution are essential input data for numerous operational applications in meteorology and hydrology, such as quantitative precipitation estimation (QPE), nowcasting, real-time initial conditions for numerical weather prediction, hydrological modelling, etc. Incorrect values may affect the results of these applications; this applies especially to unreasonably high or false zero precipitation values.

In recent decades, the number of automated weather station networks providing measurements with high temporal resolutions (e.g. 1-, 5-, or 10-minute) has rapidly increased. Consequently, procedures for data quality control (QC) have developed from manual or semiautomatic to fully automatic checks that provide relevant quality information, such as quality flags or quality indices (Lewis et al., 2021). However, in the case of precipitation, the effectiveness of automatic quality control methods has been proven to be much lower than in the case of other meteorological parameters (You et al., 2007). The key issue is the spatiotemporal variability of the precipitation field, which can be very intermittent and small-scale, depends strongly on the type of precipitation (e.g. convective or frontal), and also depends on topographic variables in mountainous areas with complex terrain (Scherrer et al., 2011).

This paper presents the RainGaugeQC scheme with automatic QC procedures, developed at the Institute of Meteorology and Water Management – National Research Institute (IMGW), which operates the Polish national meteorological and hydrological service. The scheme focuses on telemetric rain gauge measurements, and is designed to identify erroneous or suspicious data and to assign a quality index to the individual measurements.



### 1.1 Sources of errors in rain gauge data

Ground rain gauge measurements, like other observations, are affected by different types of errors, usually classified as random, systematic and gross errors. Random errors vary in an unpredictable manner, while systematic errors remain constant or vary in a predictable way, and can often be reduced. Gross errors are characterised by rare occurrence and large magnitude (WMO-No. 488, 2017).

Problems relating to the accuracy of precipitation measurement have been well documented (e.g. Sevruk, 1996; Habib et al., 2001; Golz et. al., 2005; Sieck et al., 2007; Sevruk et al., 2009). The magnitude of measurement errors depends on many factors, including weather conditions at the collector, the location of the rain gauge, and the gauge type. The most significant measurement errors are related to wind (Sevruk et al., 2009; Rasmussen et al., 2012; Martinaitis et al., 2015). Wind-induced losses mainly depend on wind speed and turbulence, as well as the type of precipitation (e.g. rain, mixed snow and rain, or snow). The measurement error is usually greater for solid than for liquid precipitation (WMO-No. 8, 2018). Because of slow falling, snow hydrometers are more susceptible to deflection by wind-induced turbulence around the gauge, making snowfall measurements prone to large systematic errors (Rasmussen et al., 2012). In windy conditions, the underestimation of snowfall accumulation frequently ranges from 20% to 50% or even higher, and additionally depends on other variables, such as exposure and the type of rain gauge (Rasmussen et al., 2012; Buisán et al., 2017; Grossi et al., 2017). Other systematic error sources are related to physical processes, such as evaporation from a bucket, wetting, and splashing. All such errors are typically referred to as catching losses.

Additional difficulties occur in winter precipitation measurements as a result of snow collecting on the gauge or snow accumulating within wind shields, either of which can completely or partially block the gauge orifice (Goodison et al., 1998; Rasmussen et al., 2012; Martinaitis et al., 2015; Kochendorfer et al., 2020). In consequence, Martinaitis et al. (2015) identified a secondary but important impact from gauges that had become partially or completely stuck during winter precipitation events. Thawing due to increased surface ambient temperatures resulted in gauges reporting false non-zero precipitation after having collected solid precipitation. These impacts became increasingly complex when rainfall occurred simultaneously with the thawing of accumulated solid precipitation.

Moreover, the accuracy of precipitation measurements may be affected by improper exposure of the gauge, site altitude, shielding or obstacles (e.g. trees, buildings) near the rain gauge, the impact of topographic variables in complex areas, and the seeder–feeder effect (Førland et al., 1996; Sevruk and Nevenic, 1998).

Additionally, mechanical problems specific to each type of rain gauge influence the accuracy of precipitation measurements. Tipping bucket rain gauges are subject to random errors related to partial or total blockages of the mechanism due to accumulated mineral or biological particulates: dust, insects, blown grass, etc. (Sevruk, 1996; Upton and Rahimi, 2003). In consequence, even partial clogging of the gauge can result in erroneous estimates of the intensity and duration of rainfall. Another specific problem with tipping bucket rain gauges relates to high-frequency bucket tips (double tips), which lead to the recording of spurious high rainfall intensities, while on the other hand very slow tips (i.e. a limited tipping rate) may result in misleading underestimates of rain rates (Upton and Rahimi, 2003; Shedekar et al., 2016).

In the case of weighing gauges, the most relevant sampling errors are related to the response time of the measurement system and the consequent systematic delay in assessing the exact weight of the accumulated precipitation in the container, especially in the case of high resolution (e.g. a 1-minute time resolution). Sampling errors may also affect the measurement of low-intensity rain (Colli et al., 2013).

Electronic weighing precipitation gauges are less susceptible to evaporation losses than tipping bucket gauges and have better accuracy in assessing the beginning of snowfall events. A heated tipping bucket gauge starts recording with a delay due to the time needed to melt the snow and fill the first tip, and measures less precipitation due to heating-related losses (Savina et al., 2012).



Furthermore, precipitation measurements may be affected by gross errors, mainly caused by the malfunctioning of
measurement devices, or occurring during data transmission.
**1.2 Approaches to quality control of rain gauge data**
Quality control is a vital part of data processing. The World Meteorological Organisation (WMO) encourages the use of data
QC in order to achieve a certain standard for international data exchange (WMO-No. 488, 2017). The WMO recommends
initially to perform real-time basic QC of raw data at sensor level, then near-real-time QC, and finally non-real-time extended
QC (semi-automatic) at the headquarters. Performing QC at various stages of data processing makes it possible to identify the
majority of errors in the dataset.
Generally speaking, some precipitation data QC checks consider each single observation separately (Upton and Rahimi,
2003; Taylor and Loescher, 2013; Blenkinsop et al., 2017), whereas more complex ones also take into account data from
neighbouring stations (Steinacker et al., 2011; Scherrer et al., 2011) or multi-source data, such as weather radar data (Yeung
al., 2014; Baserud et al., 2020) and output from a numerical weather prediction model (Qi et al., 2016). Recently, due to the
increased utilisation of crowdsourced observations, specific QC methods applicable for this type of precipitation data have
been developed (de Vos et al., 2019; Bárdossy et al., 2021; Niu et al., 2021).
For assessing the reliability of observations, several approaches are adopted. In practice, various measures of the quality
of precipitation data are used, which indicate the reliability of individual sensors resulting from measurement precision, which
is strongly conditioned by construction and technology (Førland et al., 1996), location, current meteorological conditions
(wind, temperature), etc. Often, flags describing the quality of the data are used qualitatively; for example, the WMO
recommends a scheme of five quality flags, defined as good, inconsistent, doubtful, erroneous, and missing (WMO-No. 488,
2017, p. 201).
In the simple approach to QC outputs, the only possible result is the acceptance or rejection of particular observations.
An observation that passes all of the checks is flagged as correct. If an observation fails a check, it is flagged as incorrect and
does not undergo the remaining checks (Baserud et al., 2020); however, it is possible to retrieve information on which test was
failed for each observation. Some QC schemes integrate the results of individual QC checks to generate a final flag for each
observation. In this case an adjustment test or specially designed rule base is applied to minimise the number of correct
observations that are flagged as "erroneous" – for example, if an observation failed a climate-based range test but passed the
spatial check, then an adjustment test may reduce the severity of the flag obtained from the climate-based range check (Fiebrich
et al., 2010; Lewis et al., 2018; 2021).
In another approach, after failing specific checks the measured values are not removed, but corrected. Such a method
may be used to replace suspicious data with values obtained from interpolation data from neighbouring stations (Michelson,
2004), but it does not provide any additional information. Also, the use of data from other measurement systems is not a
satisfactory solution, as these data are generally inconsistent with each other due to the extremely different error structures.
Generally, the correction of measured values can give unreliable results due to the high level of arbitrariness.
Recently, machine learning using artificial neural networks has been employed as a tool for automated quality control
as well as for the correction of errors and reconstruction of missing values in precipitation data (Moslemi and Joksimovic,
114 2018).

Quantitative indicators based on various forms of quality indicator can also be used, describing the quality of the
measurement by means of a number, most often in the range from 0.0 (completely unreliable measurement) to 1.0 (perfect
measurement) (Einfalt et al., 2010; Szturc et al., 2022).
The latter approach is adopted in the QC scheme described in this paper. In the developed RainGaugeQC scheme, the
quality of uncertain measurements is lowered and only measurements very likely to be erroneous are removed – they are
replaced with "no data" values. The advantage of this approach is that the quality information can be very useful in further





applications, for example in quality-based spatial interpolation of rain gauge data and in merging observations from different
measurement techniques (e.g. Jurczyk et al., 2020). It seems optimal to take into account quantitative information about the
quality of individual measurements in such a way that the more uncertain data are assigned a lower weight than more reliable
data.
**1.3 Structure of the paper**
The paper is structured as follows. Section 1 provides an overview of the factors influencing the accuracy of rain gauge
measurements and the main approaches to data quality control procedures. Section 2 briefly describes the rain gauge data on
which the RainGaugeQC scheme proposed in the paper was developed and calibrated, as well as the radar data used as auxiliary
data in this scheme. In section 3, the checks that constitute the RainGaugeQC system are presented (their detailed descriptions
are included in the appendices). Section 4 presents and discusses specific examples of the scheme's performance and a general
analysis of its operation. The article ends with a list of conclusions resulting from the operational use of the RainGaugeQC
scheme at IMGW (section 5).
**2 Data sources**
**2.1 Rain gauge network in Poland**
The Polish national meteorological and hydrological service, provided by IMGW, operates a nationwide meteorological
telemetric network which consists of 503 rain gauges equipped mainly with tipping bucket sensors (Fig. 1). At the synoptic
stations, SEBA Hydrometrie (https://www.seba-hydrometrie.com/) RG-50 devices are installed, whereas lower-level stations
use mainly the Met One Instruments (https://metone.com/) 60030 and 60030H devices (unheated and heated, respectively).
Telemetric precipitation measurements are available with a 10-minute time resolution: all year round for heated sensors, and
in the warm part of the year – from April to October – for unheated ones.

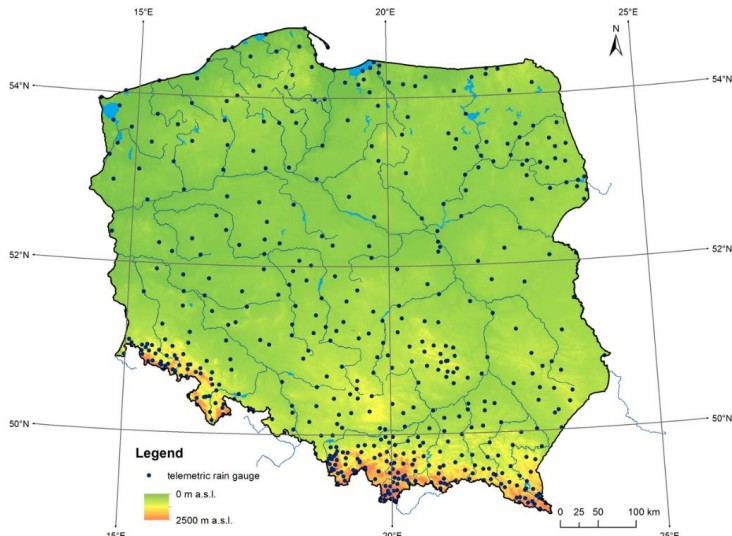

**Figure 1: Network of telemetric rain gauges in Poland.**
The reliability of individual rain gauges depends on the type of the gauge and its location, and changes with time. The
network's tipping bucket devices often malfunction, and moreover these sensors lower the precipitation values by an average
of about 8–20% (Urban and Strug, 2021).





Fig. 2 shows the relationships between measurements of 10-minute precipitation accumulations from unheated and
heated sensors on two sample rain gauges: in Dzierżoniów, located in the foothills area, during July 2021 (left), and in Nowa
Wieś Podgórna, located in the lowland Wielkopolska (Greater Poland) region in central Poland, during June 2021 (right). Both
gauges are equipped with tipping bucket devices. The correlation coefficient calculated for pairs of values in which at least
one is different from zero is extremely high for Dzierżoniów, being equal to 0.997 (Fig. 2a), while for Nowa Wieś Podgórna
it is only 0.694 (Fig. 2b), a fairly low result caused by very large differences between the values measured simultaneously by
the two sensors at the same location. The reason for such low correlation may be that tipping bucket gauges are susceptible to
frequent sensor failures.
Generally, the left graph of Fig. 2 corresponds to a well-functioning rain gauge, and the right graph to a rain gauge
providing data with large errors. For the latter, one or both sensors recorded erroneous precipitation values, and they therefore
require effective quality control. It is shown in section 4.3, concerning an example case study, how the quality control scheme
presented in this paper worked on these obviously incorrect measurements.

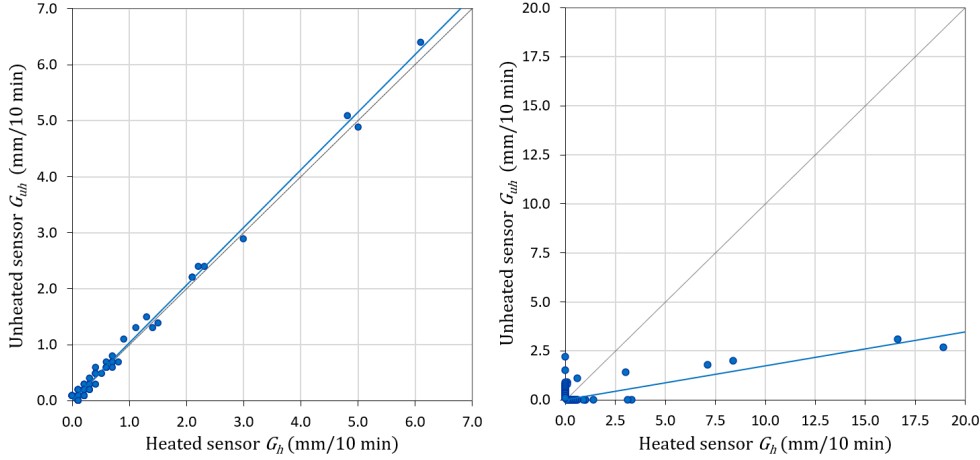

**Figure 2: Relationships between observations of 10-minute precipitation accumulations measured with tipping bucket rain gauges**
**equipped with two sensors – unheated and heated – in Dzierżoniów during July 2021 (left) and in Nowa Wieś Podgórna during June**
**2021 (right). The blue lines mark the trends of these relationships.**
**2.2 Weather radar data**
Weather radar data are employed in the RainGaugeQC scheme as auxiliary data to verify rain gauge observations. They are
generated by the Polish radar network POLRAD, which consists of eight C-band Doppler radars from Leonardo Germany
GmbH (formerly Gematronik and Selex) (Szturc et al., 2018). Three of them are dual-polarisation radars, and the others will
be upgraded to that functionality in the near future. Three- and two-dimensional radar products are generated by Rainbow 5
software every 10 min, with a 1 km spatial resolution within a 215 km range. The Marshall–Palmer formula is used to transform
the reflectivity values measured by radar into the precipitation rate, this being the most common form of such a relationship
(Neuper and Ehret, 2019). The data are quality controlled by the dedicated RADVOL-QC system developed at IMGW
(Ośródka et al., 2014; Ośródka and Szturc, 2022).

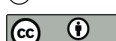



**3 General description of the developed quality control scheme**
**3.1 Set of RainGaugeQC algorithms**
A shortened version of the description of the algorithms used in the scheme was presented in works by Otop et al. (2018) and
Jurczyk et al. (2020). This section and the related appendices provide a full description of the developed algorithms. All
parameters defined here were optimised for 10-minute precipitation accumulations (mm/10 min).

**Table 1. List of sequential checks for precipitation QC.**

| ID | Abbreviation | Name | Main approach | Result of the check |
|---|---|---|---|---|
| 1 | GEC | Gross Error Check | Detection of exceedance of the natural limit | Removal of incorrect values |
| 2 | RC | Range Check | Detection of exceedance of climate-based threshold at an individual gauge | $QI$ reduction for suspiciously high precipitation value |
| 3 | RCC | Radar Conformity Check | Checking of the conformity of rain gauge and radar observations | Removal of false "no precipitation" data. For false precipitation reports, $QI$ reduction depending on $SF(G_h, G_{uh})$ and location |
| 4 | TCC | Temporal Consistency Check | Checking of the consistency of time series from heated and unheated sensors | $QI$ reduction for inconsistent sensors |
| 5 | SCC | Spatial Consistency Check | Checking of the spatial consistency of adjacent gauges | $QI$ reduction for outliers depending on the inconsistency level |


The rain gauge quality control procedure developed at IMGW consists of several checks (Table 1). Firstly, simple
plausibility tests – the gross error check and range check – are performed on a single measurement; then more complex checks
are performed, using data from both measurement sensors at the site and data from weather radars.
Before the checks, each sensor is assigned the perfect $QI$ value (1.0). In case of failure of a particular check, the $QI$
value is decreased by a specified value. If the final $QI$ value (after all of the checks) is very weak ($\leq 0.0$), the sensor is
considered useless and the measurement value is replaced with "no data".
The sensor which obtained a higher final quality index is used for further applications, but if both sensors are of the
same quality, then the heated sensor is taken.
**3.2 Similarity function (SF)**
It is useful to introduce a tool to check the similarity of two sums of precipitation. For this purpose a similarity function ($SF$)
has been proposed and is used in some of the checks. The function comparing data from two sensors $G_h$ and $G_{uh}$ (heated and
unheated) installed at a given rain gauge location, $SF(G_h, G_{uh})$, in order to check whether the precipitation values are
consistent, is defined as follows:
If ($G_h < 1.0$ mm or $G_{uh} < 1.0$ mm), then
$\qquad$ if ($|G_h - G_{uh}| < 1.0$ mm), then $SF(G_h, G_{uh}) =$ "true" $\hfill (1)$
$\qquad$ else $SF(G_h, G_{uh}) =$ "false"
whereas:
If ($G_h \geq 1.0$ mm and $G_{uh} \geq 1.0$ mm), then
$\qquad$ if $\left( 0.5 < \frac{G_h}{G_{uh}} < 2.0 \text{ or } |G_h - G_{uh}| < 1.0 \text{ mm} \right)$, then $SF(G_h, G_{uh}) =$ "true" $\hfill (2)$
$\qquad$ else $SF(G_h, G_{uh}) =$ "false"
In the above formulae, precipitation units are given in "mm", but they may refer to different accumulation periods, for
example, mm per 10 minutes (mm/10 min) or 1 hour.





The result of the use of *SF* to assess the similarity of measurements between two sensors (heated and unheated) in rain
gauges is presented in Fig 3. The graph shows example data for one day, 22 May 2019, obtained from all measuring stations.
It is indicated which measurements from the two sensors are shown by the *SF* function to be similar (marked blue) and which
are not similar (marked brown). The two blue dashed lines delimit the area in which the values measured by the unheated and
heated sensors are similar according to the *SF* function.

**Figure 3: Precipitation data from $G_{uh}$ and $G_h$ sensors that are similar (blue) and not similar (brown). The similarity of the**
**measurements from all rain gauges on 22 May 2019 was determined using the similarity function SF. The two dashed lines delimit**
**the area in which the measurements are considered similar.**
**3.3 Gross Error Check (GEC)**
GEC is a preliminary check to identify gross errors, mainly caused by the malfunctioning of measurement devices or by
mistakes occurring during data transmission or processing (Steinacker et al., 2011), which have a strong effect on the further
analyses. GEC examines whether the rain gauge measurement is within the physically acceptable range limits: not less than 0
mm and not above 56 mm/10 min. The upper limit was determined on the basis of a formula developed to estimate the
maximum reliable precipitation for various durations in Poland (Burszta-Adamiak et al., 2019). A measurement that fails the
check is rejected from further processing.
**3.4 Range Check (RC)**
RC verifies a single measurement against a threshold value, which is based on local climatological data with respect to seasonal
variation of observations in the specific location of the rain gauge. This test identifies data as implausible when they exceed
the expected maximum value, that is, the threshold empirically estimated from long-term climatological data. It is essential to
ensure reliable values of the threshold, because, for example, too low a threshold may cause extreme values of precipitation to
fail the test (Taylor and Loescher, 2013). Therefore, Fiebrich et al. (2010) recommend developing regionally specific
thresholds for the test. In the proposed QC procedure, the thresholds were defined as 10-minute precipitation values with a 1%
probability of being exceeded, determined separately for warm and cold seasons. These values were calculated for each
telemetric gauge, based on the statistical distribution of 10-minute accumulations over a long time. In the case that the
examined measurement exceeds the relevant threshold value, it is treated as suspicious and its *QI* is reduced by 0.25.



**3.5 Radar Conformity Check (RCC)**

RCC is performed to identify false precipitation – false zero and false gauge-reported precipitation measurements – on the basis of radar data, which quite reliably indicate the spatial distribution of precipitation. RCC compares each precipitation observation lower than 0.2 mm/10 min with radar observations at the gauge location and in a surrounding grid of 3 pixels x 3 pixels (the pixel size is 1 km x 1 km). If the radar data for the vicinity of the gauge are above a predefined threshold, then a "no precipitation" result measured by the sensor is assumed to be false and the *QI* is reduced to 0.

On the other hand, the RCC compares every precipitation observation with radar observations at the gauge location and in a neighbouring grid of 3 pixels x 3 pixels. If the radar data indicate "no precipitation" (0 mm), with radar data quality above a predefined threshold, then the precipitation measured by the sensor is assumed to be false and the *QI* of that observation is reduced. The reduction depends on whether data are available from one or two sensors, on their similarity, and on the gauge location (in mountains, foothills, or lowland areas).

For a detailed description of the RCC algorithm and the criteria for determining the reduction of *QI*, see Appendix 1.

**3.6 Temporal Consistency Check (TCC)**

This check, in the form described below, is possible only when two sensors are installed at each measuring station, most often heated and unheated, as is currently the case in the IMGW network. If this is not the case, then a method commonly used in quality control of various meteorological quantities is checking of the time continuity of the measured values. For some types of meteorological data the time consistency checks are efficient; however, in the case of precipitation data, this check would eliminate not only all questionable data but also a large amount of true data, in particular extreme values, because of the high variability of precipitation (WMO-No. 305, 1993, p. VI.21, VI.23).

A preliminary check is performed to detect a clogged sensor, which occurs if the same value is repeated over a certain period of time. In this case, the sensor's quality is reduced to 0.

In the next step, pairs of rain gauge sensors ($G_h$, $G_{uh}$) are tested for the existence of large differences between them. This check requires measurements from both rain gauge sensors at the same location, and can thus be conducted only in the warm half of the year. In this procedure, if the number of measurement pairs is sufficient, they are accumulated and their similarity is checked using the SF function (see section 3.2). If the sums differ, the data from both sensors have failed the TCC check and their quality is reduced.

For a detailed description of the TCC algorithm see Appendix 2.

**3.7 Spatial Consistency Check (SCC)**

SCC is applied to identify outliers based on a comparison with neighbouring gauges. Additionally, radar data are introduced to assess the level of *QI* reduction for outliers.

There are several steps in the operational procedure for SCC. Firstly, the domain area is divided into basic subdomains with a spatial resolution of 100 km x 100 km. For each subdomain, a set of percentiles of rain gauge data and the median absolute deviation (*MAD*) are calculated.

The criterion for the spatial consistency of an individual sensor is implemented based on the index *D*, calculated using the formula of Kondragunta and Shrestha (2006). The index is compared with the threshold values defined by its set of percentiles, making it possible to determine the different classes of outliers. The check is repeated for subdomains obtained by making shifts of 25 km in all four directions. If the sensor value is identified as an outlier in the basic subdomain and in the shifted subdomains, the sensor is detected as an outlier and a further procedure is applied to assess the relevant quality reduction.




For each detected outlier, two criteria are checked: (i) if data from both sensors are available for a given rain gauge and
they are similar, i.e. $SF$ ($G_h$, $G_{uh}$) = "true", and (ii) if the data passed the TCC test, then the $QI$ for the sensor is not reduced.
Otherwise, for additional verification, radar data in a grid of 5 x 5 pixels around the gauge location are considered if they are
of good quality; then the reduction of the $QI$ value depends on the class of the outlier (weak, medium, or strong) and the
magnitude of the disparity with the radar data.
A detailed description of the SCC algorithm and the criteria for reduction of the $QI$ value are given in Appendix 3.
The check may optionally analyse data from both sensors together or separately, and may or may not include data from
the previous time step. It was investigated how these settings influence the performance of the check.
Fig. 4 presents graphs showing the percentage of data with reduced $QI$ values, as a result of analysing the spatial
conformity of data from two types of sensors (unheated and heated) separately or together. The obtained sample results
generally showed large variation; however, the numbers of strong outliers increased significantly (about 2.35% versus 0.6%)
when the two types of sensors were analysed separately – in that case the algorithm appears much less tolerant.

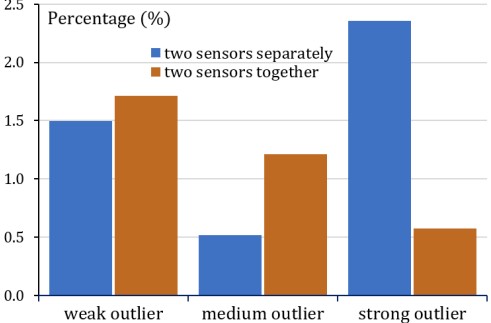

**Figure 4: Percentage of classes of outliers (weak, medium, and strong) when analysing the data from two types of sensors (unheated**
**and heated) separately (blue) or together (brown). Data from 22 May 2019.**
If the algorithm takes into account data not only from the current time step, but also from 10 minutes ago (both sensors
analysed together), then these numbers are slightly higher for weak and medium outliers and slightly lower for strong ones.
The percentage of the data belonging to different classes of outliers was slightly over 3% (Fig. 5), and for particular classes
ranged from about 1.5–1.7% for weak to about 0.6–0.9% for strong outliers. This observation suggests that the inclusion of
data from the previous time step makes the algorithm more tolerant.

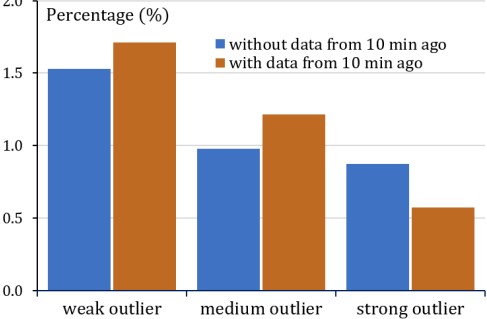

**Figure 5: Percentage of classes of outliers (weak, medium, and strong) when analysing measurements from the given time only (blue)**
**and also from the previous time step (brown). Data from two days: 20–21 June 2020.**





In the RainGaugeQC scheme used by IMGW in real time, in the SCC check both types of sensors are analysed together,
also taking account of the data from the previous time step.

### 3.8 Quality index of spatially distributed rain gauge data

In most applications of rain gauge data, spatial interpolation of the point data is required, which can be performed using one
of the many commonly known methods. However, it is not enough to spatially interpolate the $QI$ values assigned to individual
rain gauges, but it is also necessary to take into account the fact that due to the natural high variability of the precipitation field,
the uncertainty of the estimated field decreases very quickly with increasing distance from the nearest rain gauge. Therefore,
the quality field for the spatially distributed precipitation data depends on two factors: the $QI$ point values for individual rain
gauges (denoted by the $QI$ with the index "$p$") and a factor that depends linearly on the distance from the nearest rain gauge
(with the index "$d$").
The $QI$ point values from rain gauges should be spatially interpolated by the same method as the precipitation field is
interpolated; hence the quality field $QI(G_{\text{int}}(x,y))_p$ is obtained. In the case of the operational scheme used by IMGW, ordinary
kriging is applied, where the domain of 900 km x 800 km is divided into 16 subdomains of 225 km x 200 km and interpolation
is performed separately in each of them.
The factor related to the distance from the rain gauges $QI(G_{\text{int}}(x,y))_d$ takes into account the decrease in the quality of
the rainfall field depending on the distance $d(x,y)$ to the nearest rain gauge. The distance factor for each pixel is calculated
from the linear formula:
$$QI(G_{int}(x,y))_d = \frac{d_{max} - d(x,y)}{d_{max}}$$    (3)
where $d_{\max}$ is the limit value of the distance to the nearest rain gauge, above which the quality at that pixel is assigned a value
of zero (the adopted limit is 100 km).
The field of the final quality index for the rain gauge-based precipitation field is calculated from the product of the two
above factors:
$$QI(G_{\text{int}}(x,y)) = QI(G_{\text{int}}(x,y))_p \cdot QI(G_{\text{int}}(x,y))_d$$    (4)

### 4 Examples of QC scheme operation for a rain gauge with low quality measurement

### 4.1 Influence of differences in values from two sensors on precipitation field estimation

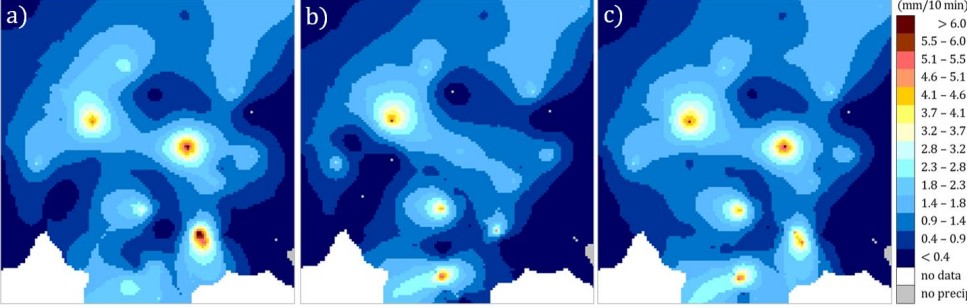

**Figure 6: Spatially interpolated rain gauge data obtained from: (a) unheated and (b) heated sensors, and (c) after quality control (considered optimal). Data from 5 August 2021, 17:40 UTC, excerpt from the Polish domain (240 km x 250 km).**

In the example presented in Fig. 6, it can be seen that the data from the two sensors can sometimes be significantly different.
The final rainfall field can be simply generated by taking the mean or the higher values of the two sensors at the same location,





and both of these approaches can be justified depending on the final application of the data. The approach used in the
RainGaugeQC scheme makes it possible to choose the better value according to defined checks, and moreover to apply it along
with the relevant *QI* value in quality-based interpolation algorithms which generate the optimal rain gauge field.

### 4.2 Result of the performance of the QC scheme after the introduction of erroneous values

Fig. 7 illustrates the performance of the proposed QC scheme. If the rain gauge data are not subjected to QC algorithms, then
two alternative data sets can be considered: from unheated (Fig. 7a) and heated (Fig. 7b) sensors. The third diagram shows an
example of data disturbed with an artificial value of 10 mm/10 min at the heated sensor of the Siercza rain gauge (Fig. 7c), the
location of which is marked with a red circle in all diagrams.

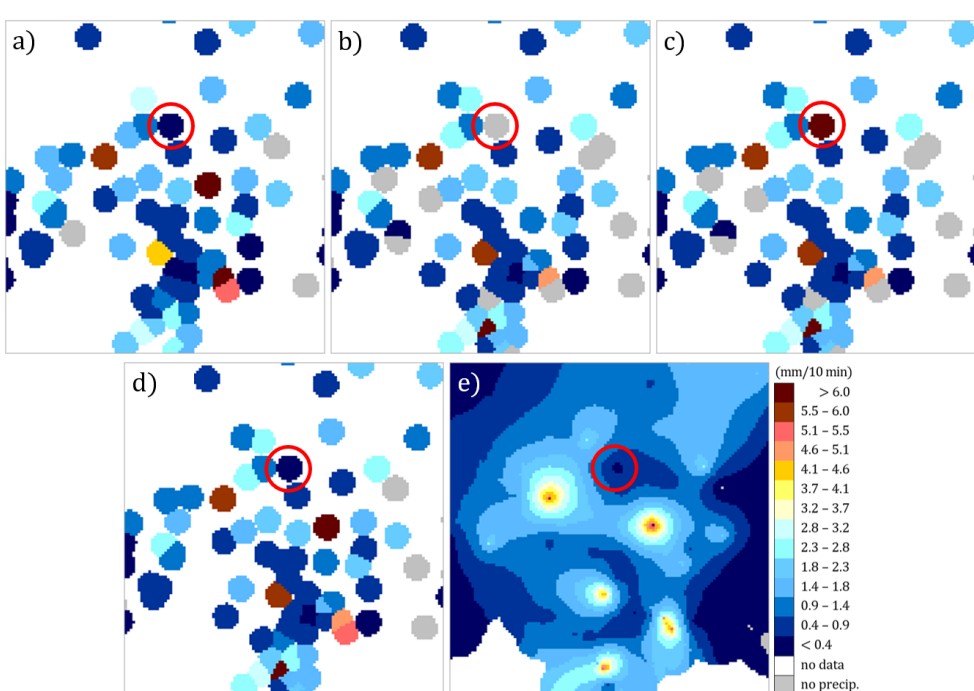

**Fig. 7. Example of the RainGaugeQC performance after the introduction of erroneous precipitation value: (a) original rain gauge**
**data from unheated sensors ($G_{uh}$) (in all fields the Siercza rain gauge is marked with a red circle), (b) original data from heated**
**sensors ($G_h$), (c) data from heated sensors disturbed with an artificial value at Siercza (10 mm/10 min), (d) rain gauge data after**
**quality control, and (e) after spatial interpolation. Data from 5 August, 2021, 17:40 UTC, excerpt from the Polish domain (240 km**
**x 250 km).**
Fig. 7d shows the values from individual rain gauges after quality control, and Fig. 7e shows the same values after
spatial interpolation using the ordinary kriging technique (this field is identical to the one shown in Fig. 6c). As these images
show, the precipitation values obtained after data quality control are some mixture of those data from both sensors that passed
the QC with higher *QI* (see section 3.1). The Siercza rain gauge, marked with a red circle, serves here as an example of a gauge
with incorrect measurement (the original values were 0.2 and 0.0 mm/10 min for unheated and heated sensors, respectively).
The erroneous value of 10 mm/10 min was eliminated as a result of the QC algorithms, so the rainfall value for this rain gauge
after QC is 0.2 mm/10 min measured by the unheated sensor.



### 4.3 Example for Nowa Wieś Podgórna rain gauge from 22 June 2021, 13:30 UTC

An example of a rain gauge with low-quality measurements, taken from the Nowa Wieś Podgórna rain gauge during June 2021, is shown in Fig. 2b (section 2.1). The low quality is evidenced by large differences between the values measured with heated and unheated sensors: the heated sensor recorded much higher 10-minute precipitation accumulations than the unheated one. The data from 22 June 2021, 13:30 UTC are analysed in detail below. The heated sensor of the Nowa Wieś Podgórna rain gauge reported a very high rainfall of 18.9 mm/10 min, whereas the unheated one reported only 2.7 mm/10 min (Table 2). If QC is not performed, then the heated sensor is generally considered the primary sensor as it operates all year round. The precipitation field resulting from the interpolation of rain gauge data without QC obtained by the ordinary kriging method is shown in Fig. 8a.

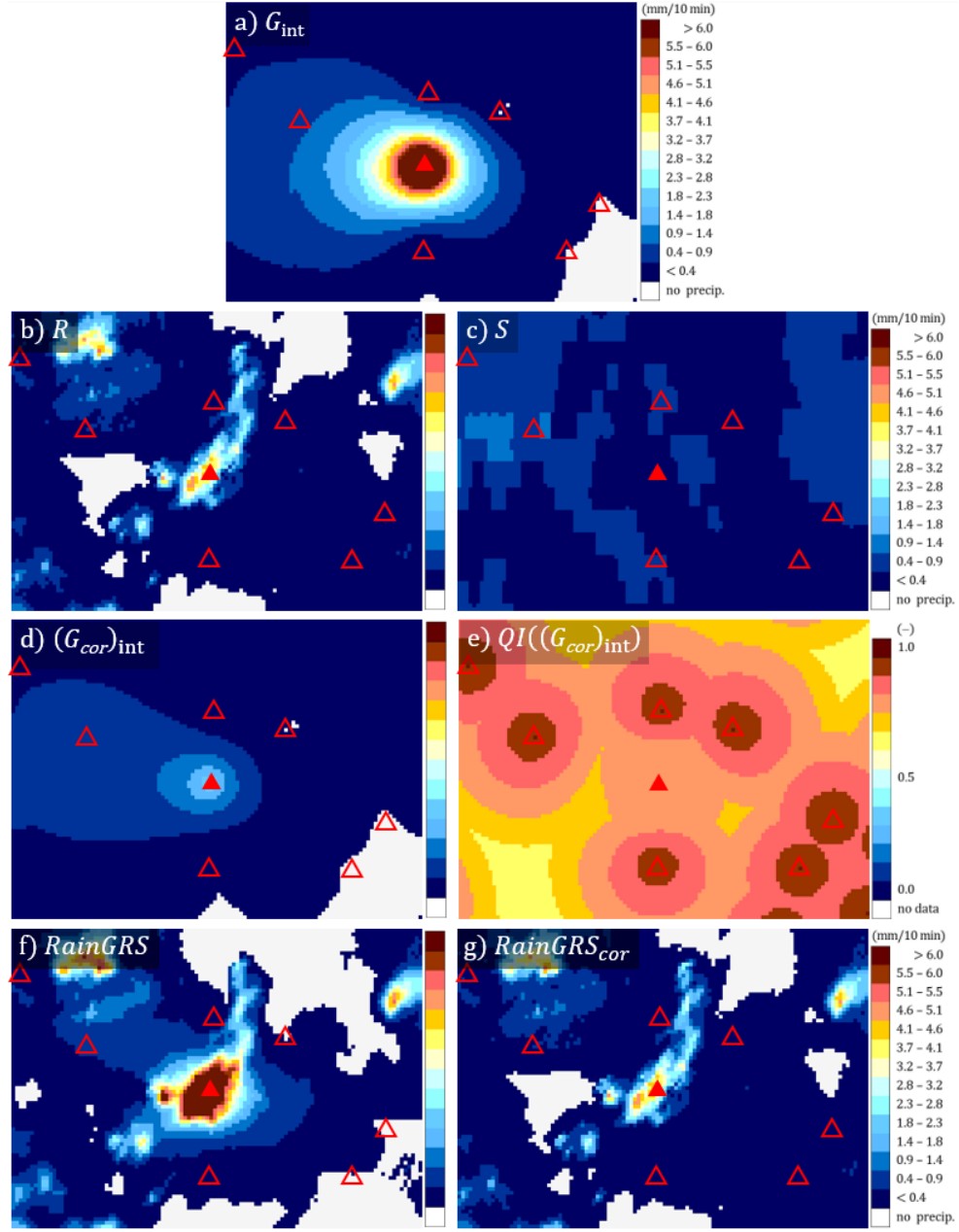

**Figure 8: Various fields of 10-minute precipitation accumulation (in mm/10 min) in the vicinity of the Nowa Wieś Podgórna rain gauge (marked with a red triangle; the locations of other rain gauges are marked with empty triangles): a) spatially interpolated field from rain gauge data without QC ($G_{int}$), b) radar-based precipitation field ($R$), c) satellite-based precipitation field ($S$), d) spatially interpolated field from rain gauge data after QC (($G_{cor}$)$_{int}$), e) $QI$ field for the precipitation field from rain gauge data after QC ($QI(($G_{cor}$)_{int})$), f) multi-source precipitation field ($RainGRS$) obtained from raw rain gauge data, g) multi-source precipitation field ($RainGRS_{cor}$) obtained from rain gauge data after QC. Data from 22 June 2021, 13:30 UTC, excerpt from the Polish domain (110 km x 80 km).**

In order to diagnose the large difference between the two sensors, a detailed investigation of the situation was performed based on precipitation data from other sources. The radar composite map from the SRI (surface rainfall intensity) product showed 3.95 mm/10 min at this location (Fig. 8b), which is much closer to the value from the unheated sensor. Satellite rainfall, determined from various NWC-SAF products based on Meteosat data (Jurczyk et al., 2020), showed only 0.05 mm/10 min





(Fig. 8c); however, measurements based on data from visible and infrared channels are much less accurate than radar
measurements. Thus, the radar data confirmed that the rainfall that occurred in the analysed time step in the close vicinity of
this rain gauge is significantly higher than in the surroundings, but not by as much as the heated sensor reported – it is much
closer to the observation of the unheated sensor.
Visually, this conclusion seems to be unquestionable, but it may be interesting how the designed RainGaugeQC scheme
functioned in this situation.

**Table 2. Results of QC of the Nowa Wieś Podgórna rain gauge station on 22 June 2021, 13:30 UTC.**

| Sensor | $G$ (mm/10 min) | Check | | | | $QI$ ($G$) (–) |
|---|---|---|---|---|---|---|
| | | RC | RSC | TCC | SCC | |
| Unheated | 2.7 | Passed | Passed | Failed | Weak outlier | 0.75 |
| Heated | 18.9 | Passed | Passed | Failed | Strong outlier | 0.50 |


The quality of the data from this rain gauge was 0.75 for the $G_{uh}$ sensor and 0.50 for $G_h$. This difference in $QI$ values
was a result of the SCC test, which showed that the $G_{uh}$ sensor differs slightly, and the $G_h$ sensor differs significantly, from the
rainfall values in the neighbouring rain gauges within the given subdomain. At the same time, both sensors failed the TCC
test, which in turn indicates that the accumulated values measured by these two sensors over the last 12 time steps differ
significantly (Table 2). This also contributed to a reduction in the final $QI$ value.
Thus, finally, the value from the unheated sensor $G_{uh}$ is taken for further processing. The precipitation field after the
spatial interpolation of QC data obtained by the ordinary kriging method is shown in Fig. 8d. The precipitation values around
this rain gauge location are clearly lower than those shown in Fig. 8a (without QC). The $QI$ field for spatially interpolated rain
gauges is shown in Fig. 8e – the Nowa Wieś Podgórna rain gauge is of lower quality than the neighbouring rain gauges.
QC of rain gauge data influences the precipitation fields produced by applications for the generation of multi-source
fields. This is shown by the example of the QPE fields produced by the RainGRS system, which operationally combines
precipitation data from rain gauges, weather radar and meteorological satellites, based on conditional merging and additionally
taking quality information into account (Jurczyk et al., 2020). In Fig. 8 two fields generated by RainGRS are presented: based
on rain gauge data without QC and after QC (Figs. 8f and 8g, respectively). Applying quality controlled rain gauge data, the
RainGRS estimate decreases from 16.19 to 3.26 mm/10 min, which is a very significant effect.
**4.4 General effects of the operation of the scheme**
The performance of the RainGaugeQC scheme can be analysed in terms of the degree of $QI$ reduction. This is presented in
Fig. 9, for individual months representative for autumn, winter, spring and summer conditions (October, January, April, and
July, respectively).





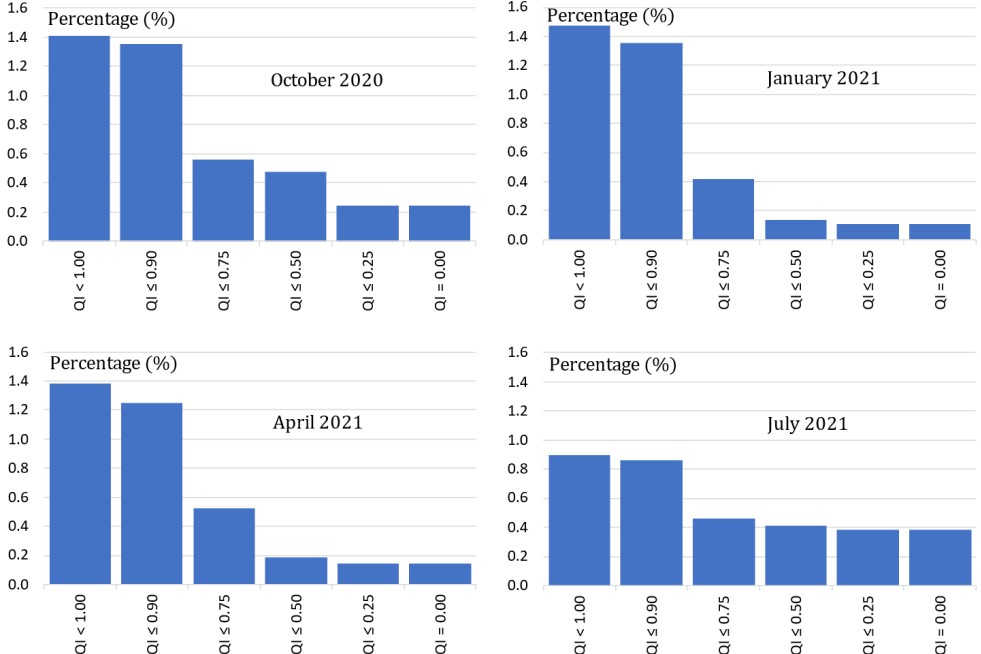

**Figure 9: Percentage of rain gauge observations with a specified *QI* reduction after quality control. From top: percentage contribution in each *QI* interval; cumulative percentage contribution (in %); from left: October 2020, January, April, and July 2021.**

The graphs shown do not include the percentage contribution of measurements that were assigned a quality of 1.0; this is equal to about 92.0–94.5%, many times higher than the total contribution of all other values. In general, it can be seen from Fig. 9 that by far the greatest number of reductions in *QI* values was to values in the range [0.75, 0.90), and this is observed in all seasons of the year. Relatively large numbers of *QI* reductions to values in the range [0.50, 0.75) occur in winter (January) and spring (April), and relatively many reductions to a zero value occur in summer (July) and autumn (October).

The number of rain gauge observations with reduced quality is relatively small, below 1.5%. For example, the contribution of data with *QI* reduced to zero ranges from about one-third to one-tenth, but grows to about one-half over the summer (July). In practice, this means that these data were rejected. Probably the most important reason is that in the summer there often occurs convective precipitation, characterised by high intensities and strong spatial variability, and moreover rain gauges in no-rain situations react to morning dew condensation, which gives false rainfall measurements sometimes as high as 0.3 mm/10 min.

The most diversified distribution of *QI* reductions is observed in winter (January): most often there are small decreases in the *QI* value. In summer (July), this distribution is the least varied, which can be partially explained by the numerous *QI* reductions to zero.

**5 Conclusions**

1.  Quality control of rain gauge data is essential, especially from the perspective of operational applications, when it is not possible to verify gauge data employing highly reliable precipitation measurements, such as manual Hellmann rain gauges, which are not available in real time.

2.  It seems that the RainGaugeQC approach to the QC of rain gauge data, which consists in estimating the value of the *QI* of individual observations, enables more effective use of the data. On the one hand, it is a more cautious approach,





as it does not eliminate all suspicious observations, and on the other hand, it enables flexible treatment of any
suspected case of data incorrectness.
3.  The IMGW rain gauge network consists mostly of rain gauges equipped with two sensors: unheated and heated. This
unique equipment allows the use of pairs of data to conduct much more effective QC. Comparing the observations
from two sensors installed at the same location significantly increases the possibility of obtaining information about
the uncertainty of measurements, for example by checking the time consistency of the data (TCC check). This is
especially important when measurements are carried out with tipping bucket rain gauges, which have relatively low
reliability. The availability of observations from both sensors is especially important during the warm season, when
convective phenomena prevail. The frequent lack of two sensors installed at the same location reduces the scheme's
effectiveness to some extent; however, it remains at a satisfactory level.
4.  It is worth considering the possibility of employing radar data in the RCC and SCC algorithms to detect erroneous
rain gauge measurements and to assess their reliability, based on the difference between the values from rain gauge
and weather radar. The case study proved that the RainGaugeQC system can identify regionally inconsistent data
thanks to the use of radar data as well as neighbouring rain gauge data.
5.  The presented set of algorithms is based on empirical relationships that are strongly dependent on local conditions,
both technical and geographic. The most important factors are the density of the rain gauge network, the availability
of other data that can be used as a reference for QC (e.g. from the weather radar network), the type of sensors (their
failure rate and measurement uncertainty), as well as terrain orography, wind conditions, and surface precipitation
type. Therefore, any changes in the network configuration necessitate recalibration of the algorithms.
6.  The number of rain gauge observations with reduced $QI$ following QC under the RainGaugeQC scheme is relatively
small, as it is below 1.5%. In all seasons, the highest number of $QI$ value reductions was to values in the range [0.75,
0.90). The highest number of erroneous data (with $QI$ reduced to zero) is found in summer (July) (approximately
0.4%), whereas in other seasons it ranges from about 0.10% to 0.23%.
**Appendix 1. Detailed description of the Radar Conformity Check (RCC) algorithm**
RCC is performed to identify false zero precipitation and false gauge-reported precipitation measurement by applying radar
data.
1.  Identifying false zero precipitation.
Each gauge sensor value ($G$) less than 0.2 mm/10 min is checked against radar observations ($R$) at the gauge location
and in its vicinity within a grid of 3 pixels x 3 pixels.
If at least one pixel of radar data had precipitation above 0.4 mm, then the gauge value measured by this
sensor is assumed to be erroneous, thus the sensor value is replaced by "no data" and the quality of this sensor is
reduced to 0.
2.  Identifying false gauge-reported precipitation.
Each gauge sensor value ($G$) above 0 mm/10 min is checked against radar observations ($R$) at the gauge location and
in its vicinity within a grid of 3 pixels x 3 pixels.
If at least two radar pixels with $QI > 0.85$ returned "no precipitation" ($R = 0$ mm/10 min), then the following
conditions are checked:
(a) If for a given rain gauge, data are available only from one sensor ($G$) and $G > 0$ mm/10 min, then:
– if the gauge is located in a mountain or foothill area, the sensor is considered erroneous and its value is
replaced by $G = 0$ mm and its quality reduced by 0.5;





469   –   if the gauge is located in a lowland area, the sensor is considered erroneous and its value is replaced by
470    $G = 0$ mm and its quality reduced by 0.25.

471   (b)   If for a given rain gauge, data are available from two sensors (heated $G_h$ and unheated $G_{uh}$) and $G_h > 0$ mm/10
472    min and $G_{uh} > 0$ mm/10 min, then:

473    –   if the gauge is located in a mountain or foothill area and values from both sensors are similar, i.e. $SF$ ($G_{uh}$,
474     $G_h$) = "true", then the quality of both sensors is reduced by 0.75, but if $SF$ ($G_{uh}$, $G_h$) = "false" then their
475     qualities are reduced to $QI = 0$ and the sensor values are replaced by "no data";

476    –   if the gauge is located in a lowland area, then the sensor qualities are reduced to $QI = 0$ and the sensor
477     values are replaced by "no data".

478   (c)   If for a given rain gauge, data are available from two sensors (heated $G_h$ and unheated $G_{uh}$) and one of them
479    reports "no precipitation" (i.e. $G_h = 0$ mm/10 min or $G_{uh} = 0$ mm/10 min), then:

480    –   if the rain gauge is located in a mountain or foothill area and the values from both sensors are similar (i.e.
481     $SF$ ($G_{uh}$, $G_h$) = "true"), then the $QI$ of the sensor which observed precipitation $G > 0$ mm/10 min is reduced
482     by 0.75, but if $SF$ ($G_{uh}$, $G_h$) = "false", then the $QI$ of the sensor which reports $G > 0$ mm/10 min is reduced
483     to $QI = 0$ and the sensor value is replaced by "no data";

484    –   if the rain gauge is located in a lowland area, then the quality of the sensor that reports $G > 0$ mm/10 min
485     is reduced to $QI = 0$ and the sensor value is replaced by "no data".

486   **Appendix 2. Detailed description of the Temporal Conformity Check (TCC) algorithm**

487   A preliminary check is performed to detect constant values. If the same value (e.g. 0.1 mm/10 min) is reported for a certain
488   number of time steps (e.g. nine consecutive observations), then the sensor is probably clogged. In this case, the blocked sensor
489   has failed the TCC test, its $QI$ is reduced to 0, and the TCC test cannot be performed for the other sensor.

490    The main part of TCC serves to identify pairs of rain gauge sensors ($G_h$, $G_{uh}$) for which there are large differences
491   between simultaneously measured values, which may be evidence of their low quality. This check requires measurements from
492   both rain gauge sensors at the same location; it can thus be conducted only in the warm season, when both sensors provide
493   measurements. This lasts from April to October, when data from unheated sensors ($G_{uh}$) are available; the heated sensors ($G_h$)
494   operate all year round.

495   1.   Pairs of simultaneous measurements from two sensors are verified for the last 12 time steps, but observations with
496    poor quality are not taken into account. If the number of quality-verified pairs (for previous time steps with $QI > 0.0$,
497    and for the current one passing the previous checks, i.e. GEC, RC and RCC) is high enough (at least 9), the
498    cumulative sums are calculated:

499    $$S_h = \sum_{i=1}^{n} G_{h,i}, \quad S_{uh} = \sum_{i=1}^{n} G_{uh,i} \tag{5}$$

500   2.   The similarity of the accumulated sums is checked by means of the $SF$ function. If they differ significantly, i.e. if
501    $SF(S_h, S_{uh})$ = "false", then the data from both sensors have failed the TCC test and their quality is reduced by 0.25.

502   **Appendix 3. Detailed description of the Spatial Consistency Check (SCC) algorithm**

503   The SCC procedure consists of the following steps:

504   1.   The Polish domain (900 km x 800 km) is divided into subdomains with dimensions of 100 x 100 km. Only data
505    with $QI > 0$ after previous tests are subject to this check. Both sensors, heated and unheated, can be analysed
506    together or separately, and these data can also be analysed together with data from the previous time step (10 min

ago) if their $QI = 1.0$. In order to perform this check, the number of data in a subdomain must be at least three;
otherwise the test is not performed for that subdomain.
2.   Based on data from rain gauges ($G$) located in a given subdomain, the following percentiles are determined: 25%,
50% (median), and 75% ($Q_{25}(G)$, $Q_{med}(G)$, and $Q_{75}(G)$).
The median absolute deviation ($MAD$) for a given subdomain is determined from the formula:
$$MAD = \frac{1}{n}\sum_{i=1}^{n}|G_i - Q_{med}(G)| \tag{6}$$

where $n$ is the number of data, $G_i$ is the $i$-th sensor value, and $Q_{med}(G)$ is the median.
3.   The index $D_i$, which determines numerically the deviation of the precipitation value measured with the $i$-th sensor
from the values of sensors within a given subdomain, is calculated from the formula (Kondragunta and Shrestha,

2006):

$$D_i = \begin{cases} 0 & MAD = 0 \\ \frac{|G_i - Q_{med}|}{MAD} & MAD \neq 0 \text{ and } Q_{75} = Q_{25} \\ \frac{|G_i - Q_{med}|}{Q_{75} - Q_{25}} & MAD \neq 0 \text{ and } Q_{75} \neq Q_{25} \end{cases} \tag{7}$$

Following calculation of the $D_i$ values for all sensors within a given subdomain, three percentiles are
determined: 90%, 95%, and 99% ($Q_{90}$, $Q_{95}$, and $Q_{99}$).
4.   If $D_i \leq Q_{90}(D)$, then the $i$-th sensor is not an outlier and the test is passed.
If this is not the case, the $i$-th sensor is flagged and the formula (8) is applied to compare the index $D_i$ with
the three percentile values, in order to determine to which class of outliers the given value belongs:
$$\text{outlier} = \begin{cases} \text{strong} & D_i > Q_{99}(D) \\ \text{medium} & Q_{95}(D) < D_i \leq Q_{99}(D) \\ \text{weak} & Q_{90}(D) < D_i \leq Q_{95}(D) \end{cases} \tag{8}$$

The procedure is repeated in four subdomains resulting from shifting the given subdomain vertically and
horizontally, i.e. in four directions, with offsets of 25 km (except for subdomains on the edges and corners of the
domain, which are shifted in three and two directions, respectively). If the value measured with a given sensor is
flagged in all analysed subdomains, it fails the SCC check. If the values belonged to different classes of outliers,
the weakest one is assigned to the sensor for further processing.
5.   For sensors that failed the SCC check, if the data from both sensors are available for a given rain gauge and they
are similar, i.e. $SF(G_h, G_{uh}) =$ "true", and passed the TCC check, then the $QI$ for the sensor is not reduced.
Otherwise, each outlier is verified against radar data. For this purpose the following values are determined
within a grid of 5 pixels x 5 pixels around this rain gauge location: $\min(QI(R))$ – the minimum quality $QI$ of the
radar precipitation $R$; $R_{max} = \max(R: QI(R) > 0.75)$ – the maximum value of radar precipitation with a quality
above 0.75; $QI(R_{max})$ – the quality of the maximum value of radar precipitation $R_{max}$. This verification algorithm
is as follows:
If $\min(QI(R)) > 0.75$, then: $\hspace{4cm}$ (9)
if $R_{max} = 0$, then the quality is reduced by 1.0 and $G =$ "no data";
if ($G > 1.0$ mm) and $\left(\frac{G}{R_{max}} < \frac{QI(R_{max})}{4.0} \text{ or } \frac{G}{R_{max}} > \frac{4.0}{QI(R_{max})}\right)$, then:
$$QI = \begin{cases} QI - 1.00 & \text{strong outlier} \\ QI - 0.50 & \text{medium outlier} \\ QI - 0.20 & \text{weak outlier} \end{cases}$$

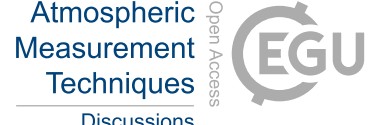

if $(G > 1.0 \text{ mm})$ and $\left( \frac{G}{R_{max}} \geq \frac{QI(R_{max})}{4.0} \text{ and } \frac{G}{R_{max}} \leq \frac{4.0}{QI(R_{max})} \right)$, then:
$QI = \begin{cases} QI - 0.25 & \text{strong outlier} \\ QI - 0.10 & \text{medium outlier} \\ QI & \text{weak outlier} \end{cases}$
If $(G \leq 1.0 \text{ mm})$ or $\left( \min(QI(R)) \leq 0.75 \right)$, then:
$QI = \begin{cases} QI - 0.25 & \text{strong outlier} \\ QI - 0.10 & \text{medium outlier} \\ QI & \text{weak outlier} \end{cases}$
A simpler analysis of the spatial consistency of rain gauge data may be performed (especially if radar data are
unavailable), analogously to steps 1–4, but with only the $Q_{95}$ percentile being determined. Here, if in all subdomains $D_i \leq$
$Q_{95}(D)$, the sensor fails the SCC, and the $QI$ is decreased by 0.10.

*Author contributions.* KO, IO, and JS designed algorithms of the RainGaugeQC system. KO developed the software code and
performed the simulations. JS, IO, and KO prepared the manuscript. JS made figures.

*Competing interests.* The authors declare that they have no conflict of interest.

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
