# Peer review of "Automatic quality control of telemetric rain gauge data providing quantitative quality information (RainGaugeQC)"

_Atmospheric Measurement Techniques, 2022_

## Author Comment (AC1)

**Replies to comments from Reviewer # 1**

**General comments:**

- There are sometimes logically strange places. Some of them are because of lack of explanations.

- It is difficult to read some sentences due to their complicated structures.

- For the reviewer, the usage of the term "rain gauge" is incongruous in this manuscript. The reviewer thinks the authors should use the word "station" to show one observation site.

- The reviewer recommends asking for the help of a professional English editor. The manuscript should be carefully revised by the authors before submitted to an English editing company.

> We would like to thank the Reviewer for devoting a lot of time to a very thorough analysis of our article and for numerous, very valuable comments! We tried to take into account all comments on the details of the described algorithms. We also tried to revise the text applying general remarks concerning the clarity and logic, and then consulted it with a native speaker. We hope that it significantly improved the quality of our article.

**Specific comments:**

Abstract:
The abstract lacks the description of analyses evaluating the quality check method, for example, evaluation using horizontally gridded distribution of precipitation.

> We have inserted a new sentence in l. 15:

> "The performance of this scheme has been evaluated by analysing the spatial distribution of the precipitation field and comparing it with precipitation observations and estimates provided by other techniques. The effectiveness of the RainGaugeQC scheme was analysed in terms of the statistics of QI reduction."

l. 33:
Is this "the RainGaugeQC scheme" a name as a proper noun, in other words, the name of a specific QC scheme?

> We have modified this sentence a bit:

> "This paper presents the RainGaugeQC software, which is a package of automatic QC procedures, …"

l. 103:
It seems that this "correct" is unnecessary.

We think that in the context of the whole sentence the adjective "correct" is needed to point out that the procedure should prevent flagging correct observations as "erroneous".

l. 116:
Do you mean a (numeral) value by this "a number"? I think this expression is inappropriate for an academic report.

We have modified this sentence:

„describing the quality of the observations expressed in numbers, most often …"

l. 118:
What do you mean by "The latter approach"? The approach described in lines 115–117? In this description, "the former approach" is unclear.

We merged these two paragraphs, and the first sentence of the previous second paragraph begins with the words:

„This approach…

ll. 126–127:
I think that this sentence is not necessary because readers have already read Section 1.

Of course, the reviewer is right. We deleted that sentence, and now the second sentence of the paragraph starts with the words:

„After section 1 ….."

ll. 149–150:
The positions of Dzierżoniów and Nowa Wieś Podgórna should be shown in the Fig. 2 for foreign readers.

Thank you for this remark. We have added the figure with location of Dzierżoniów, Nowa Wieś Podgórna, and Siercza stations.

ll. 154–155:
What are reasons for much smaller values of precipitation in the unheated sensor than in the heated sensor, brought from the difference in the type of sensors? If the reason can be judged as a malfunction in the specific sensors, the replacement of the sensors should be prioritized.

We agree that one of these two sensors should be replaced. However the investigation of each of them separately does not indicate a malfunction: symptoms of clogging or blockage of one of the sensors are not evident. Only after several days of observing the time series and calculating the correlation coefficient, it can be concluded that there is something wrong in the measurements, and then the service can take action. Therefore, the detection of such faults is not easy for the service and only the introduction of QC makes it easier.

l. 168:

Where are these radars? I suggest to the authors to briefly show the radars' functions and positions in a table. The positions can be plotted in Fig. 1.

> We have added locations of the radar sites to Fig. 1., however, we have not included the table with parameters, because all radars of this network have practically the same parameters (except for the dual-pol measurement).

l. 170:
Do you mean dual-polarization by "that functionality"? The part after ", and the others" seems to be unnecessary for the present paper.

> We changed this sentence to:
>
> „… radars, and work is currently underway on upgrading all the radars, including dual polarization functionality."

ll. 189–190:
Why do you give priority to heated sensors?

> We gave priority to the heated sensor, because it operates all year round, which ensures better time continuity of measurements.

l. 193:
Please clearly give the definition of the value $G$.

> We have added the definition of the G value (l. 193-195):
>
> „… two sensors $G_h$ and $G_{uh}$ (heated and unheated) installed at the same gauge station $G$, $SF(G_h, G_{uh})$, …"

ll. 229–230:
I think that this criterion is too strict. For example, for a 30-years period, ~16,000 measurements (correspond to ~110 days) are subject to a QI reduction by 0.25.

> We assumed that the uncertainty of the precipitation measurement (expressed by the dimensionless quantity "QI") increases with the intensity of the precipitation, especially for values above the defined threshold, so we decided that such an indirect reduction of QI is justified. Nevertheless, we are considering increasing this threshold, if our tipping-bucket rain gauges are upgraded.

l. 231:
Please show a specific analysis period for determining the thresholds.

> We added the information in l. 231:
>
> "…10-minute accumulations in a 30-year time series (1986-2015)."

ll. 235–237:
Is the RCC conducted when a gauge measures precipitation < 0.2 mm/10 min? If so, this sentence seems to be incorrect.

We have changed this sentence:

"RCC compares each gauge observation lower than 0.2 mm/10 min with radar observations at the gauge location and its surrounding of 3 pixels x 3 pixels (the pixel size is 1 km x 1 km)."

ll. 239–240:
Is this sentence repetition of the previous paragraph?

In paragraph l. 234-238 we remove erroneous observations $G < 0.2$ mm / 10 min, while in paragraph l. 239-243 we remove all erroneous measurements of precipitation $G > 0$ when the radar measurements have a sufficiently high quality QI(R). Indeed, it was not clearly explained, so we improved this second paragraph:

„On the other hand, the RCC compares every sensor observation $G > 0$ mm/10 min with radar observations at the gauge location and its neighbouring of 3 pixels x 3 pixels. If the radar data is of a quality $QI(R)$ above a predefined threshold and indicates "no precipitation" (0 mm), then the precipitation measured by the sensor is assumed to be false and the $QI$ of that observation is reduced. The reduction depends on whether data are available from one or two sensors, on their similarity, and on the gauge location (in mountains, foothills, or lowland areas)."

ll. 240–242:
A criterion for gauge measurement is not shown here.

See the previous answer.

ll. 247–248:
I cannot understand why you use the present progressive form here.

Thank you for pointing us to this mistake, of course we have corrected it.

ll. 254–258:
Why did you include the check in this paragraph in the temporal consistency check?

The SF function is applied in two checks: TCC and SCC. In the TCC it investigates the similarity of data series from two sensors (heated and unheated) of a given station, in order to check whether the two series are consistent.

ll. 255–256:
I can guess that the check is influenced by snow, but I cannot understand the reason why the check is unable in the cold half of the year.

This is due to the lack of availability of a second time series. Only heated sensors measure precipitation all year round, because unheated ones do not work in winter when snowfall often occurs (in Poland from October to April). During this period, two time series from the same station are not available, so it is not possible to compare them. We have added an explanation to l. 256:

"…half of the year, because only then two time series from the same station are available."

l. 291:
The sentence is unclear especially what the value "over 3%" indicates.

> We corrected this sentence:
>
> "The percentage of the data belonging to **all** classes of outliers **together** was slightly over 3%".

ll. 292–293:
I think that this is only in the cases with strong outliers. If so, you should clearly describe it. Or you should specify the reason why you can say the check is tolerant judging from the results in all the 3 categories of outliers.

> These numbers refer to the individual classes of outliers. We have corrected this sentence to make it understandable:
>
> „… and for particular classes **varies** from about 1.5–1.7% for **the** weak to about 0.6–0.9% for **the** strong outliers."

l. 305:
I think the more distant from the nearest gauge, the more uncertainty the interpolated precipitation data has.

> Thank you for this remark. It was a mistake in edition the manuscript. It should be:
>
> "… the uncertainty of the estimated field increases very quickly with **increasing** distance from the nearest rain gauge".

ll. 309–310:
What is the reason why the QI values should be spatially interpolated as the precipitation is interpolated? How do you calculate the QI( $G_{int}$( x, y ))$_p$?

> We need the QI values in each pixel of the precipitation field for multi-source precipitation estimation. Both interpolations (of G and QI values) are performed simultaneously, using the same parameters, so in both cases there is the same contributions from the individual rain gauges. In this way, both fields are completely consistent.
>
> We have changed the sentence (l. 309-310):
>
> "The precipitation and *QI* point values from rain station are spatially interpolated simultaneously by the same method using the same parameters, so in both cases there is the same contributions from the individual rain gauges. Hence the obtained quality field $QI(G_{int}(x,y))_p$ is completely consistent with precipitation field $(G_{int}(x,y))$."

l. 329:

Did you describe that a larger value is employed from the precipitation observation by heated and unheated gauge sensors? What is the reason justifying it? Is this against what written in ll. 189–190?

>We have listed these two possible simple approaches as options that can be used, however we do not use them as we employ the quality criterion (QI). The higher value approach is sometimes used operationally (e.g. at IMGW) applying these data as input to the rainfall-runoff models. It gives the user more confidence that no high rainfall will be missed. We have clarified this in the text:

>„**In simpler solutions** the final rainfall field can be generated by taking the mean or the higher values of the two sensors at the same location, and both of these approaches can be justified depending on the final application of the data."

l. 347:
This description seems to include an error. An interpolated precipitation field is not a distribution of observed precipitation with gauges.

>Thank you for this remark. We changed this sentence as follow:

>"Fig. 7d shows the values from individual rain gauges after quality control, and Fig. 7e shows the precipitation field after spatial interpolation using the ordinary kriging technique (this field is identical to the one shown in Fig. 6c)."

l. 370:
What do you mean by this "excerpt"?

>It means that the fig. 7 shows only part (rectangle with dimensions 240 km x 250 km) of the whole Polish domain. We changed this word to "fragment".

l. 375:
Please spell out of the NWC-SAF.

>We have added:

>"NWC SAF (Satellite Application Facilities on Support to Nowcasting and Very Short Range Forecasting)"

l. 377:
What do you mean by this "Thus"? The reviewer feels that the logical connection with the previous sentence is strange.

>We have removed "thus".

l. 395:
Please spell out the RainGRS.

>RainGRS system is a tool for the estimation of a high-resolution precipitation field, developed and implemented at the IMGW. It is a proper name of the system, not an acronym.

Fig. 9:
Is this analysis include periods without precipitation?

You should give exact labels showing the ranges of QI for the horizontal axes (e.g. [0.90, 1.00) for the left most bar).

> This analysis includes 0 mm values as well.

> Values do not apply to ranges, but to values below the given QI value. Hence, successive bars give lower and lower values.

l. 410:
Are these values about 98.5–99.1% estimated from the shown histograms?

> Thank you for pointing out a mistake! In the text we have listed incorrect values which have been corrected: 98.5 – 99.1%.

"many times" can be written with "much".

> Thanks, it has been corrected.

l. 411:
This part can be written as "between the ranges of (0.50, 0.75] and (0.75, 0.90]".
The value range [0.75, 0.90) is different from that in the labels for the horizontal axes in Fig. 9. It will be "(0.75, 0.90]".

> Thank you for pointing out the inconsistency between the graph (in Fig. 9) and the description in the text. We changed the marking of closed and open ranges (square and round brackets) to the correct ones.

l. 413:
What do you mean by "relatively many reductions to a zero value"?

> We changed to:

> "… relatively many data has a quality that is reduced to zero …"

l. 420:
What do you mean by this "diversified distribution"?

> "The most diverse distribution of QI reduction …"

l. 455–460 (item 1):
In this rule, the case with precipitation of 0.1 mm / 10 min will be regarded as an error if the radar detects the precipitation over 0.4 mm (/ 10 min ?). The reviewer thinks that this rule is too strict.

This sub-algorithm is primarily intended for the detection of clogged rain gauges, which is rather common in the tipping-bucket devices. Hence, we made this really quite restrictive condition.

l. 458:
Is this value for 10 minutes?

Yes, it is 0.4 mm/10 min. We fixed it.

ll. 461–485 (item 2):
I think the authors should note that the rule is stricter in mountainous or foothill areas.

Thank you for this remark. The information that the rule depends on location in mountainous, foothill or lowlands areas there is in ll. 242-243.

l. 464:
This rule makes the case, with (1) similar values (for example, meeting the SF criterion) in the gauge observations and radar observation at a pixel including the gauge station, (2) no precipitation in the radar observation at the other pixels, will be classified as an errorneous data. I feel this is unreasonable.

We agree with the reviewer's observation. This only occurs when the quality of the radar data is very high (QI > 0.85). In such a situation, however, we trust the radar data more.

l. 467:
How do you define mountainous, foothill or lowland areas?

We use the height above sea level as a criteria for define lowland, foothill, and mountainous areas:

"lowlands: areas below 300 m a.s.l.; foothills: between 300 and 600 m a.s.l. ; mountainous areas: above 600 m a.s.l.".

This information has been supplemented in l. 243.

l. 495:
How long is this time step?

All calculations in our paper are carried out in 10-minute time step (see Section 3.1, l. 179).

ll. 495–496:
How do you define observations with poor quality?

This is explained in the second sentence of paragraph II. 495-498.

l. 507:
Do you mean the number of gauge stations by this lower-limit number of "three"?

We have corrected the sentence:

"… there must be data available from at least three stations in a subdomain …"

ll. 509–512:
What do you define the values of the median and 25, 75 percentiles for? What are the values G and $Q_{xx}$?

We have improved the description:

"$(Q_{25}(G), Q_{med}(G), \text{and } Q_{75}(G))$" into "$(Q_{25}(G), Q_{med}(G), \text{and } Q_{75}(G), \text{respectively})$"

Consequently, we have also supplemented the abbreviations of the percentiles of various quantities, e.g.: $Q_{25}(G)$ and $Q_{25}(D)$.

l. 515:
"the values of sensors within a given subdomain" is not specifically defined.

We have corrected:

"The index $D_i$, which determines numerically the deviation of the precipitation value measured with the i-th sensor from the median of all sensors within a given subdomain …."

l. 517:
What cases do you assume for the case when $Q_{75}$ equals $Q_{25}$?

Such situations are possible only when the precipitation values at all stations differ very little.

ll. 518–519:
I think that you can determine $Q_{90}$, $Q_{95}$ and $Q_{99}$ values without $D_i$.

In l. 519 we calculate the percentiles not of G (rainfall), but of the variable Di (Eq. 7). Indeed, the abbreviations we had introduced are confusing – we have revised them throughout Appendix 3.

l. 520:
In this procedure, don't you test beyond the border of the domains but only in the shifted subdomains within one domain?

We do not move subdomains beyond the domain – the algorithm takes this limitation into account, as described in ll. 524-526.

l. 524:
What do you mean by "vertically"? I think the domains and the subdomains are defined in the horizontal area.

Yes, but the subdomain is shifted in four directions i.e. vertically **(west-east)** and horizontally **(south-north)**. We have specified it in the text.

ll. 532–533:
There was no definition for the quality index of radar observation.

Indeed, there is no definition. We added it to section 2.2, in which we write about radar data (l. 174) (the reference was already there):

„The system also generates quality fields, QI(R), based on analyses of particular errors disturbing radar data."

l. 537:
Is this procedure for cases with no precipitation from gauge observation?

No, it is a step of procedure to verify outlier **sensor** data against radar data.

l. 538:
Here, you compare the values with different properties; $G / R_{max}$ is a ratio of precipitation amount, and QI/4.0 is calculated from the quality index. How do you justify such a comparison? (for example, empirically justified?)

Yes, it was determined empirically. We have added this explanation in the text in ll. 276:

"… data (the limitation imposed on the magnitude of this disparity has been determined empirically)."

**The following parts are unclear:**

l. 110:
"due to ... structures"

This sentence has been changed:

"Also, the use of data from other measurement systems is not a satisfactory solution, as these data are generally inconsistent with each other due to their different spatial distributions".

ll. 303–305:
The sentence is unclear due to a strange structure of it. Please check it again.

This text fragment has been changed:

"However, it is not enough to spatially interpolate the QI values assigned to individual rain gauges. It is also necessary to take into account the fact that the uncertainty of the estimated field increases very quickly with increasing distance from the nearest rain gauge".

l. 415:
Do you mean the cases when QI is zero? I think that this expression is unclear.

> We have improved the text. We mean the cases when QC procedure reduces quality
> index (QI) to zero:

> "….zero (i.e. $QI = 0.0$) ranges…"

l. 487

> The sentence has been modified:

> "The first step of this check is performed to detect constant values **observed by a
> given sensor**".

ll. 490–491

> The sentence has been modified:

> "The main part of TCC serves to identify rain stations for which there are large
> differences between values measured simultaneously by pairs of rain sensors ($Gh$,
> $Guh$), which may be evidence of their low quality".

ll. 505–507

> The sentence l. 505-507 has been modified:

> "It is optional: (i) to analyse both sensors, heated and unheated, together or separately,
> (ii) to include also data from the previous time step (10 min ago) if their $QI = 1.0$".

---

## Author Comment (AC2)

**Replies to comments from Reviewer #2**

**Suggestions:**

Line 63. Please expand with a half-sentence on how the concept of seeder-feeder is related to under- or over-estimation of surface rain fall amount, or how the seeder-feeder process causes a lag of surface precipitation relative to the radar observing enhanced precipitation aloft. This reviewer is not familiar with how the seeder-feeder ice cloud process impacts surface rain accumulations. Thus, I believe an AMT reader would be interested in what information the authors want to share.

> We have added an explanation:
>
> "the seeder–feeder effect" → "the seeder–feeder effect (**when precipitation from an upper-level cloud falls through a lower-level orographic stratus cloud capping a small mountain**)".

Line 112-124. Make these three paragraphs one paragraph because the phrase "The latter approach…" should refer to an approach that is in the same paragraph. Also, the first two paragraphs are one sentence each.

> Thank you for your suggestion - we have put these paragraphs together.

Line 137. Does the phrase "lower-level" refer to lower-altitude stations, or lower-quality (less reliable) stations? Please clarify.

> They are stations of a lower category ("order", not "level" - we named them incorrectly), i.e. mainly "precipitation stations" (https://glossary.ametsoc.org/wiki/Precipitation_station). We improved the terminology and changed this sentence as suggested:
>
> "…devices are installed, whereas lower-level stations use mainly…"
> → "…devices are installed, whereas **precipitation** stations use mainly…"

Lines 150-151. The sentence starting with "Both gauges…" is confusing. Should the sentence read: "Both stations are equipped with heated and unheated sensors"? Please clarify.

> We have changed this sentence as follows to be consistent with the surrounding sentences:
>
> "Both gauges are equipped with tipping bucket devices"
> → "Both **stations** are equipped with **two** tipping bucket **sensors**"

Line 156. The first sentence is confusing to me. Should the sentence be something like, "...corresponds to well-functioning rain gauges, and the right graph corresponds to one or both rain gauges not functioning correctly." Please clarify.

> We agree with the Reviewer that this fragment was unclear. We have changed these two sentences as follows:

"Generally, the left graph of Fig. 2 corresponds to a well-functioning rain gauge, and the right graph to a rain gauge providing data with large errors. For the latter, one or both sensors recorded erroneous precipitation values, and they therefore require effective quality control."
→ "Generally, the left graph of Fig. 2 corresponds to well-functioning rain **station**, and the right graph **corresponds** to rain **station with** one or both sensors **not functioning correctly**, and they therefore require effective quality control."

Line 220. Can you calculate the corresponding radar reflectivity factor needed with the MP Z-R relationship to get 56 mm/10 min rain rate? That would be an interesting comparison for radar-centric AMT readers.

We counted it and added the result:
"above 56 mm/10 min" → "above 56 mm/10 min **(i.e. 51 dBZ)**"

Section 4.3. This is a good example of how the proposed method identifies a large rain gauge data value and lowers the QI score. As a visual of how the algorithms identify this outlier and reduces the QI score, can a time-series plot be included in Fig. 8 of rain rates from neighboring gauges and radar estimates over the Nowa Wies Podgorna site for the 12 samples before and a few samples after the outlier event? Also, this timeseries plot can be used to remind the AMT reader that the time-series analysis is only looking backward in time to produce a real-time QC'd product.

We have inserted the graph that the reviewer expects along with the relevant paragraph into line 380.

[Figure]

"Fig. 9 shows the recorded precipitation time series from 12 time steps (i.e. two hours) before the analysis date (13:30 UTC), and 6 time steps after this date, at Nowa Wieś Podgórna station (two sensors) and maximum values of the four neighbouring stations. These stations are located between 19 and 35 km from the analysed Nowa Wieś Podgórna station. Until the analysis date, precipitation measured by the sensors of these stations was not high, as it was up to about 1 mm/10 min, but 20 min later a significant increase in precipitation of about 6 mm/10 min was observed on both sensors of one of the nearby stations. At the analysed time-step only Nowa Wieś Podgórna station recorded a slightly higher precipitation on the heated sensor, while it was drastically higher on the unheated sensor (Table 2)."

Lines 375 and 395. The rainRGS and NWC-SAF datasets need to introduced in Section 2, which describes the datasets that are used in the study.

We have added a new section 2.3:

**2.3. Other data**
In addition, the fields of the following precipitation estimates were used for the case studies:
- satellite precipitation fields determined from various NWC-SAF (Satellite Application Facilities on Support to Nowcasting and Very Short Range Forecasting) products based on Meteosat data (Jurczyk et al., 2020),
- QPE fields produced by the RainGRS system, which operationally combines precipitation data from rain gauges, weather radar and meteorological satellites, based on conditional merging and additionally taking quality information into account (Jurczyk et al., 2020).

In the text, we have shortened the explanations and added references to the new section 2.3. Line 375:

"Satellite rainfall, determined from various NWC-SAF products based on Meteosat data (Jurczyk et al., 2020) …"
→ "Satellite rainfall, determined from various NWC-SAF products based on Meteosat data (**see Section 2.3**) …"

Lines 395-397:
"This is shown by the example of the QPE fields produced by the RainGRS system, which operationally combines precipitation data from rain gauges, weather radar and meteorological satellites, based on conditional merging and additionally taking quality information into account (Jurczyk et al., 2020)."
→ "This is shown by the example of the QPE fields produced by the RainGRS system, which operationally combines precipitation data from rain gauges, weather radar and meteorological satellites (**see Section 2.3**)."

The appendices are well written and describe the algorithms with sufficient detail that I think this reviewer and AMT readers could repeat the algorithm with their own rain gauge network. The conclusions properly state that every network needs to be calibrated to determine their own thresholds. To help promote the algorithm, can flow diagrams be provided showing the if-then-else flow of the algorithms? I am thinking big-picture diagrams with the text providing the details. (I assume the authors already have these diagrams for their conference slide-deck oral presentations.)

With these diagrams we have a problem. We have made attempts, but in our opinion it is not possible to diagram these algorithms in such a way that it is both complete and clearer than a step-by-step description in bullet points. We attach below two working versions of the diagram for the SCC algorithm: a shorter one and a longer one. In our opinion, neither of these meets expectations, even though both are somewhat simplified relative to the full description. In summary, we therefore propose not to add these diagrams.

Shorter version:

[Figure]

Longer version:

---

## Referee Report (RR1)

Automatic quality control of telemetric rain gauge data providing quantitative quality information (RainGaugeQC)

By

Katarzyna Ośródka, Irena Otop, and Jan Szturc

Overview for the second review

The reviewer pays respect to the authors' efforts in thorough revision and recognizes that the manuscript has been largely improved. The reviewer thinks that the manuscript will almost meet the criterion for publication in Atmospheric Measurement Techniques. However, there are 3 parts with room for improvement, listed below:

Minor comments:

ll. 119–121 (old 1. 116 in my previous review):
I still feel a sense of incongruity in the phrase "in numbers". I found (missed in the first review) that you have already mentioned that these are "quantitative" indicators, so you can omit "expressed in numbers".

ll. 206–208 (old l. 193)
There is still no definition for G (Gh and Guh); I wanted to say that you should explain that G shows precipitation (amount).

Many readers can guess that the G (Gh and Guh) means "precipitation amount" observed by the heated and the unheated gauges, respectively, at the same station. However, you should clearly give a definition for all the variables appearing in the paper.

ll. 524–527 (old ll. 495–496)
I confirmed a paragraph in lines 524–527 in the new manuscript and I understood what you mean. However, it may be difficult for readers to understand it from this sentence structure. If you feel the necessity, please reconsider it.

---

## Author Response (AR2)

Referee Report:

**Overview for the second review**

The reviewer pays respect to the authors' efforts in thorough revision and recognizes that the manuscript has been largely improved. The reviewer thinks that the manuscript will almost meet the criterion for publication in Atmospheric Measurement Techniques. However, there are 3 parts with room for improvement, listed below:

**Minor comments:**

ll. 119–121 (old 1. 116 in my previous review):
I still feel a sense of incongruity in the phrase "in numbers". I found (missed in the first review) that you have already mentioned that these are "quantitative" indicators, so you can omit "expressed in numbers".

> We agree that the omission of this phrase will benefit the clarity of the sentence. We have changed:
>
> "Quantitative indicators based on various forms of quality indicator can also be used, describing the quality of the observations expressed in numbers, most often in the range from 0.0 (completely unreliable measurement) to 1.0 (perfect measurement)…"
>
> → "Quantitative indicators describing the quality of the observations can also be used, most often as a quality index ($QI$) ranging from 0.0 (completely unreliable measurement) to 1.0 (perfect measurement) …"

ll. 206–208 (old l. 193)
There is still no definition for G (Gh and Guh); I wanted to say that you should explain that G shows precipitation (amount).

Many readers can guess that the G (Gh and Guh) means "precipitation amount" observed by the heated and the unheated gauges, respectively, at the same station. However, you should clearly give a definition for all the variables appearing in the paper.

> Yes, you are right… We have added the new sentence defining these values into line 154 as these denotations appear for the first time in Fig. 2:
>
> „The following denotations are introduced for the precipitation values they measure: $G_h$ is the 10-min precipitation amount observed by the heated sensor, and $G_{uh}$ is the analogical value observed by the unheated one."

ll. 524–527 (old ll. 495–496)
I confirmed a paragraph in lines 524–527 in the new manuscript and I understood what you mean. However, it may be difficult for readers to understand it from this sentence structure. If you feel the necessity, please reconsider it.

> Indeed, after a long time we can see that this passage may be confusing. We have completely revised it:

"Pairs of simultaneous measurements from two sensors are verified for the last 12 time steps, observations with poor quality are not taken into account. If the number of quality-verified pairs (for previous time steps with $QI > 0.0$, and for the current one passing the previous checks, i.e. GEC, RC and RCC) is high enough (at least 9), the cumulative sums are calculated:"

→ "Pairs of simultaneous measurements from two sensors are verified for the last 12 time steps, excluding observations of poor quality (which $QI$ is 0.0 for previous time steps and for the current time step failed GEC, RC or RCC check). If the number of the pairs is high enough (at least 9), the cumulative sums are calculated:"

**We would like to thank the Reviewer very much once again for such insightful and precise comments and for his great patience with our paper!**